# Metrological Characterization of a CO$_2$ Laser-Based System for Inscribing Long-Period Gratings in Optical Fibers

Sebastian Valencia-Garzón [1,†], Erick Reyes-Vera [1,*,†], Jorge Galvis-Arroyave [2,†], Jose P. Montoya [1] and Nelson Gomez-Cardona [1]

1    Department of Electronics and Telecommunications, Instituto Tecnológico Metropolitano ITM, Medellín 050034, Colombia
2    Physics Metrology Division, Instituto Nacional de Metrología, Bogotá 111321, Colombia
*    Correspondence: erickreyes@itm.edu.co; Tel.: +57-304-652-9926
†    These authors contributed equally to this work.

**Abstract:** A CO$_2$ laser-based system was studied and implemented to produce asymmetric long period fiber gratings (LPFG) with a large attenuation peak, high reproducibility, and high stability. The first half of this study provides a mathematical uncertainty model of the CO$_2$ laser-based approach that takes into account various mechanical and thermal effects that impact this production technique. This is the first time that metrological analysis and modeling are performed on the CO$_2$ laser-based engraving technique. Following that, the engraved system's quality was assessed using a microscopic approach to confirm mechanical characteristics such as grating period, engraved spot width, and penetration depth, demonstrating that, if the thermal and mechanical components of the overall system are correctly managed, it is feasible to have very low inaccuracy. Lastly, the LPFG performance as temperature and strain sensors was tested, and the findings show that they had good linearity in both circumstances. Thus, the temperature sensor had a maximal sensitivity of 58 pm/°C when measuring temperature changed from 20 to 97 °C, but the strain sensor had sensitivity of 43 pm/µε when measuring strain variations from 5.59 to 25 µε. As a result, the model and results presented in this paper can be utilized to create a platform for the metrological management of lengths involved in the process of manufacturing LPFGs, devices that are widely employed in the creation of sensors and communications devices.

**Keywords:** metrology; long-period fiber grating (LPFG); CO$_2$ laser; fiber characterization; optical fiber fabrication; fiber-optic sensors

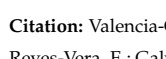



## 1. Introduction

In recent years, optical fiber has attracted considerable interest in a variety of technological fields, primarily as the physical layer of high-speed communication links, but also as the platform for various infiber optical sensing devices [1–6] . Optical fiber devices based on period gratings, such as long-period fiber gratings (LPFGs) and fiber Bragg gratings (FBGs), are among the most used because they are robust, compact, have low insertion losses, and are composed of all-fiber components. Since their introduction, LPFGs have been associated with costly manufacturing technologies, but they are now easier and less expensive to produce, gaining popularity in the optics community. An LPFG consists of periodic refractive index modulations along an optical fiber's core, thus coupling light from the fundamental core mode to a specific cladding mode [7,8]. Due to the high leakage losses of the cladding modes, an attenuation peak can be observed at specific wavelengths in the transmission spectrum of the fiber. Both the coupled mode and its corresponding transmission peak are then sensitive to changes in the temperature or mechanical bending of the cladding, and changes in the refractive index of the surrounding medium [9]. As a result, LPFGs have been used in the construction of a wide variety of telecommunication components,

such as filters, dispersion compensators, amplified spontaneous emission suppressors, and cascaded Raman amplifiers [10–13]. On the other hand, LPFGs are widely used to create sensor components that can monitor various physical and chemical parameters such as strain/stress, refractive index, and temperature [8,14,15].

The manufacturing process of an LPFG determines the final reproducibility and stability of the optical devices on the basis of the grating. As a result, numerous manufacturing processes have been researched in recent decades. Vengsarkar et al. announced the first method used to inscribe a permanent LPFG in a photosensitive single-mode optical fiber using a UV laser in 1996 [16]. Nevertheless, the UV exposure approach has significant limitations in practical applications, such as the high operational cost of UV lasers and the usage of photosensitive fibers or $H_2$-loaded fibers to increase grating writing efficiency. Other fabrication methods have lately been investigated. Some of them are electric arc discharge [17], femtosecond laser exposure [18], periodic mechanical deformation [19], chemical etching [11] and $CO_2$ laser irradiation [1,20,21]. This last alternative offers various advantages over other manufacturing processes, including having greater heating efficiency, less insertion loss, lower costs, and the used optical fiber does not require prior photosensitization. Likewise, using the well-known point-to-point approach, this technology may be regulated to produce complicated grating patterns without the use of costly masks [20]. Furthermore, $CO_2$ laser exposure incident from a side of the fiber causes an asymmetrical refractive index distribution in the cross-section of the final LPFG, resulting in a range of unique features in the LPFGs. For example, this kind of LPFGs are more sensitive to mechanical variations [21].

According to a literature review, previous research on LPFGs based on the $CO_2$ laser irradiation approach has mostly focused on the phenomena behind the manufacturing process, its applications, and the theory explaining the modal interaction [1,13,15,20–23]. For example, Davids et al. reported the first LPFG written by the $CO_2$ laser irradiation technique in a conventional glass fiber in 1998 [24] . Several variants in the method have resolved emergent issues such as vibration and repeatability control related to manufacturing faults [23,25] . Furthermore, this approach has been used to create LPFGs not just in single -mode fibers, but also in tapered and photonic-crystal fibers [12,20,26,27].

The manufacturing process of asymmetric LPFGs using the $CO_2$ laser irradiation approach is described in detail in this paper. The production setup was adjusted together with testing to ensure the process's reproducibility, stability, and traceability. As a result, to the best of our knowledge, this is the first time that a metrological investigation has been applied this extensively to the widely used point-to-point LPFG technology. The average penetration depth ($S_d$) and average spot width ($S_w$) of each engraved point were measured, and the uncertainty was determined at the *National Metrology Institute of Colombia* to confirm the mechanical characteristics of the experimental setup. Lastly, some of the manufactured LPFGs were used to measure tensility and temperature as a function of spectral behavior to give a reference sensor performance; sensitivities of 43 pm/$\mu\varepsilon$ and 58 pm/°C were obtained, respectively. A model is also presented to calculate the uncertainty associated with the LPFG manufacturing process utilizing the $CO_2$ laser engraving technology.

## 2. Materials and Methods

### 2.1. Operating Principle

Both FBGs and LPFGs work as layered inline transmission gratings that are formed by periodic variations in the refractive index on the material as is shown in Figure 1. These variations are caused by $CO_2$ laser exposure on distinct spots separated at regular intervals along an optical fiber. This configuration allows for the fundamental propagation mode in the core (in single-mode fibers) to be coupled with numerous cladding modes. These interactions depend on the grating period ($\Lambda$): below 10 μm, the structure acts as an FBG coupling light from the propagating core mode to the counter-propagating core mode, acting as a Bragg reflector that resonates in a small wavelength range and returns a narrow spectral peak, leaving a valley in the transmission spectrum. In contrast,

the period of an LPFG has a range from 10 μm to a few millimetres, coupling to different copropagating cladding leaky modes (with m = 1, 2, 3. . .) that are radiated out of the optical fiber, producing several dips or "valleys" in the transmitted spectrum. No interactions with counter-propagating modes occur in regular LPFGs. LPFG couplings satisfy a resonance condition given by the effective refractive indices of the core $n(\lambda)_{eff,co}$, the refractive index of the m-th cladding mode $n(\lambda)_{eff,cl}^{m}$, and the grating period ($\Lambda$) [28,29]. According to Equation (1), at each m-th resonance wavelength, a dip in the transmission spectrum is observed at:

$$\lambda_{res}^{i} = (n(\lambda)_{eff,co} - n(\lambda)_{eff,cl}^{i})\wedge , \tag{1}$$

where $\wedge$ is the period of the grating, $n(\lambda)_{eff,co}$ and $n(\lambda)_{eff,cl}^{i})$ are the effective indices of the fundamental core mode and $i^{th}$ cladding mode, respectively. If, during this process, the optical power of $CO_2$ laser is focused only on one side of the optical fiber, an asymmetric structure of periodic grooves is inscribed, as illustrated in Figure 2. $S_d$ and $S_w$ are the groove's depth and width, respectively.

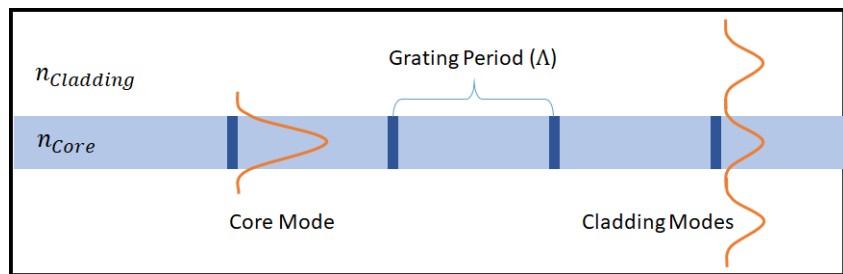

**Figure 1.** Section of the refractive index profile and transmitted modes of an LPFG inscribed in standard single-mode optical fiber; the depicted cladding modes leak out of the fiber.

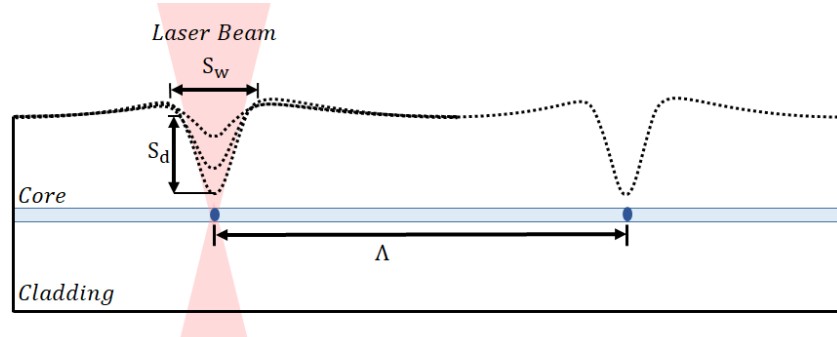

**Figure 2.** Schematic of the fiber under laser exposure during the LPFG manufacturing process. The relevant dimensions (parameters) are penetration depth ($S_d$), spot width ($S_w$), and grating period between spots ($\wedge$).

The grooves can have many geometries depending on the used method of inscription. However, to guarantee the reproducibility and repeatability of the same LPFGs, different geometrical and physical considerations must be taken: (1) the distance between the fiber optic core and the focal point of the laser beam along the inscription path; (2) the axial alignment between the $CO_2$ laser beam and the fiber; (3) the mechanical tension required to avoid lateral bends over the fiber due to thermal relaxation. As is shown in Figure 3, a polyethylene terephthalate (PET) foam placed between two fiber clamps was used to find the focal point of the $CO_2$ laser beam and track its position along the engraving path.

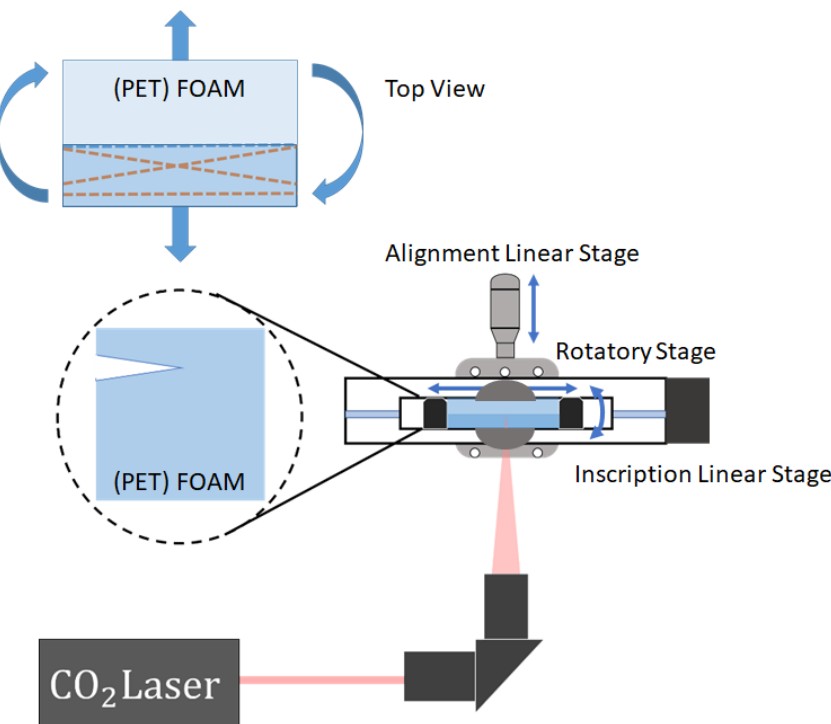

**Figure 3.** Experimental setup used for the alignment of the $CO_2$ laser on the motor travel axis. The focal point could be continuously monitored during the entire travel of the stage.

The trajectory traced in the foam by the laser was used to fix the inclination angle, and the distance between the linear stage and the focal point of the laser at each part of the stage's travel. Lastly, after mechanical adjustment and alignment, the fiber was held in place with Thorlabs HFR007 bare fiber clamps, and the long period fiber gratings were inscribed step by step by employing this new setup in a similar way to that described in [1,18].

On the other hand, the sensitivity of long-period gratings to changes as a function of temperature or strain is well-known. In fact, this sensitivity depends mainly on the grating period $\Lambda$, the excited cladding modes, and the characteristics of the optical fiber used (refractive index, thermal expansion coefficient, Young's modulus, etc.). Thus, the resonances of the LPFGs could be controlled trough the manipulation of these parameters. For this reason, the control of the manufacturing process plays an important role in ensuring traceability. This allows for the process to be repeatable, especially when the applicability of this type of sensors has been proven in different industrial environments. Likewise, it is necessary to build an uncertainty estimation model that allows for us to determine the quality of the manufacturing process.

The temperature sensitivity of a LPFG can be expressed in terms of the material contributions as the effective refractive index of the core ($n_{eff}$), the refractive index of the cladding ($n_{cl}$), their variations with respect to the temperature, and the wave-guide contributions as the grating period ($\wedge$) and length (L). Equation (2) presents the mathematical expression for the temperature sensitivity of a LPFG.

$$\frac{d\lambda}{dT} = \frac{d\lambda}{d(\delta n_{eff})}\left(\frac{dn_{eff}}{dT} - \frac{dn_{cl}}{dT}\right) + \Lambda\frac{d\lambda}{d\Lambda} \cdot \frac{1}{L}\frac{dL}{dT}, \tag{2}$$

where $\lambda$ is the central wavelength of the attenuation band, $T$ is the applied temperature, and $\delta n_{eff} = n_{eff} - n_{cl}$. The first term on the right-hand side of Equation (2) depends on the characteristics of the fiber-optic material used and is highly dependent on the order of the cladding mode. For lower-order modes of between m = 1 and m = 16, the grating period must be higher than 100 μm. For higher-order modes between m = 17 and m = 30, on the

other hand, the grating period must be lower than 100 μm. The effect of the material in Equation (2) can be dominant or negligible depending on whether the cladding modes are low- or high-order. For example, different sensors based on LPFGs with low-order cladding modes (fabricated using an SMF-28 optical fiber) reported temperature sensitivities between 30 and 100 pm/°C [10,21,30,31].

The axial strain sensitivity of an LPFG can be expressed in terms of the material contributions from the strain-optic effect and the waveguide contributions from the slope of the dispersion term and the grating period. Equation (3) presents the mathematical expression for the axial strain sensitivity of LPFGs.

$$\frac{d\lambda}{d\varepsilon} = \frac{d\lambda}{d(\delta n_{eff})}\left(\frac{dn_{eff}}{d\varepsilon} - \frac{dn_{cl}}{d\varepsilon}\right) + \Lambda\frac{d\lambda}{d\Lambda}. \tag{3}$$

Equation (3) shows that, if the LPGs have a period higher than 100 μm, the strain sensitivity is positive, and if the period is lower than 100 μm, the strain sensitivity is negative. In this work, LPFGs with a period equal to 620 μm were fabricated to evaluate the temperature and strain sensitivity of an asymmetric LPG. Then, excited three low cladding modes: $LP_{02}$, $LP_{03}$, and $LP_{04}$.

## 2.2. Fabrication of LPFGs Using $CO_2$ Laser Technique

The proposed LPFG engraving system uses a $CO_2$ laser (Infinity Series: Model 155), which has a maximal power of 50 W, a wavelength of 10.6 μm, and a repetition rate ranging from 0.5 to 100 kHz. On the other hand, a motorized linear base (ZABER Technologies, X-L-HM200A-E03) with a step resolution of 0.124 μm was used to place the optical fiber (Corning, SMF-28e), which was situated in the focal plane of the $CO_2$ laser beam. Figure 4 depicts a schematic of the experimental setup. The focal spot was obtained with a set of fixed optical components: first, a mirror, and then a Galilean beam expander composed of a plane concave, plane convex lens, and a positive meniscus, to concentrate the $CO_2$ laser beam on the fiber. The size of the spot could be regulated by varying the focal length of the meniscus lens. In our case, the engraving procedure used a spot with a diameter of 21 μm.

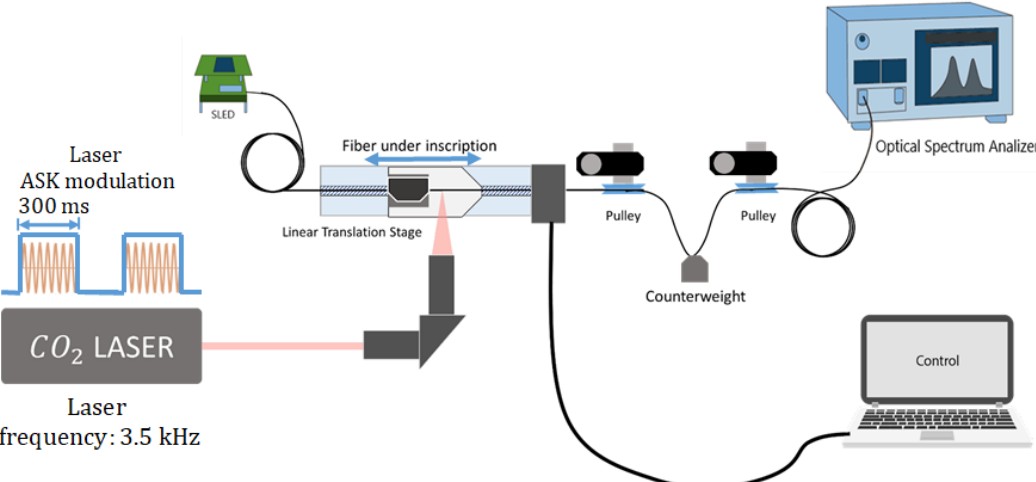

**Figure 4.** Experimental setup used to fabricate LPFGs using $CO_2$ laser technique.

The entire preparation process to fabricate the LPFGs may be structured as follows: In the first phase, the optical fiber is horizontally located directly beneath the laser beam. One end of the optical fiber is fixed using clamps for a bare optical fiber (Thorlabs, HFR007), and a permanent tension is applied to the other end using a 20 g hanging weight to avoid small bends, which also helps in the recording process being optimal. After that, the laser alignment and beam focus are tuned with the lens array in order to ensure that the beam is focused on the optical fiber and remains on the fiber throughout the entire engraving.

In this phase, parameters such as the grating period ($\wedge$), total displacement, and number of pulses are configured. The last step is to switch the laser on and irradiate the optical fiber for N pulses in one cycle. The repeated operation forms a small high-temperature zone in the fiber, resulting in $SiO_2$ gasification on the fiber surface. As a result, periodic grooves are cut into the fiber, as seen in Figure 2. Due to the photoelastic effect, these groove configurations cause the refractive index to be periodic along the fiber, resulting in LPFGs. Furthermore, an optical spectrum analyzer (Yokogawa, AQ6370B) in conjunction with an SLED light source, is utilized to examine the transmission spectrum of LPFG in real time. The spectrum indicates whether or not the LPFG is being recorded correctly. The process must be repeated multiple times until a resonant peak in the optical spectrum is produced.

*2.3. Uncertainty Model*

As previously mentioned, the main novelty of this work is the creation of a mathematical model for the characterization of the LPFG manufacturing process using the $CO_2$ laser-based technique. From Section 2, Equations (2) and (3), the grating period $\Lambda$ allows for choosing desirable characteristics for LPGs to be used as sensors. Thus, for the mathematical model, the *measurand* is the length of the grating period ($l_i$). Since the temperature affects the measurements of $l_i$, the thermal expansion coefficient of the fiber optic ($\alpha_i$) and the temperature offset $\theta_i$ have to be included into the mathematical model as is presented in Equation (4).

$$l_i(\theta_i) = l_i(1 + \alpha_i\theta_i), \tag{4}$$

Temperature offset $\theta_i$ is the difference between fiber-optic temperature $t_i$ and standard reference temperature $t_{ref} = 20\ °C$, and the coefficient of thermal expansion is $\alpha_i = 0.55 \times 10^{-6}/°C$ [32]. For the measurement of $l_i$, a digital Quick Scope was employed (Mitutoyo, QS-L2010Z/AFC). This system uses images produced by a digital microscope to establish a datum from where a coordinate measuring machine (CMM) with traceability to the International System of Units (SI) indirectly measures length. As a consequence, $l_i$ must be equal to the length measured with the Quick Scope $l_p$, as is presented in Equation (5).

$$
\begin{aligned}
l_i\ (1 + \alpha_i\theta_i) &= l_p\ (1 + \alpha_p\theta_p) \\
l_i &= l_p\ (1 + \alpha_p\ \theta_p - \alpha_i\ \theta_i - \alpha_i\theta_i\alpha_p\ \theta_p) \\
l_i &= l_p\ (1 + \alpha_p\ \theta_p - \alpha_i\ \theta_i)
\end{aligned}
\tag{5}
$$

where $\alpha_p = 11.5 \times 10^{-6}$ is the thermal expansion coefficients associated with the CMM, and $\theta_p$ is the temperature offset of the CMM, which is usually less than 0.5 °C. The factor $(\alpha_i\ \theta_i\ \alpha_p\theta_p)$ from Equation (5) is approximately zero due to the magnitude of the factors. As was suggested in the *Guide to the Expression of Uncertainty in Measurement* (GUM), the evaluation of the combined standard uncertainty $u_c$ for Equation (5) can be expressed as is presented in Equation (6):

$$u_c{}^2(x_i) = \sum_{i=1}^{N} c_i{}^2 u^2(x_i) \tag{6}$$

where terms $c_i$ are the sensitivity coefficients, which are the partial derivatives used to describe how $l_i$ varies with changes in the values of the input quantities $(\alpha_i, \theta_i, p, \alpha_p, \theta_p)$. Thus, the sensitivity coefficients $C_i$ from Equation (5) are presented in Equation (7):

$$
\begin{aligned}
C_{l_p} &= \partial l_i/\partial l_p = (1 + \alpha_p\theta_p - \alpha_i\ \theta_i\ ) \approx 1 \\
C_{\alpha_i} &= \partial l_i/\partial \alpha_i = -l_p.\theta_i \\
C_{\theta_i} &= \partial l_i/\partial \theta_i = -l_p.\alpha_i \\
C_{\alpha_p} &= \partial l_i/\partial \alpha_p = l_p.\theta_p \\
C_{\theta_p} &= \partial l_i/\partial \theta_p = l_p.\alpha_p
\end{aligned}
\tag{7}
$$

Introducing the sensitivity coefficients with their respective uncertainties, Expression (6) becomes Equation (8) for the standard combined uncertainty of the grating period length $l_i$.

$$u_c{}^2(l_i) = u^2(l_p) + (-l_p.\theta_i)^2 u^2(\alpha_i) + (-l_p.\alpha_i)^2 u^2(\theta_i) + (l_p.\theta_p) u^2(\alpha_p) + (l_p.\alpha_p)^2 u^2(\theta_p) \quad (8)$$

Since the uncertainty associated with the distance measured with the CMM $u(l_p)$ depends on the uncertainties associated with the resolution $u(res)$, repeatability $u(rep)$, calibration $u(cal)$, and maximal acceptable error $u(m)$ for the CMM, the term $u(l_p)$ must be expanded as is shown in Equation (9).

$$u^2(l_p) = u^2(res) + u^2(m) + u^2(cal) + u^2(rep) \quad (9)$$

Substituting $u(l_p)$ into Equation (8), the combined standard uncertainty $u_c{}^2$ becomes Equation (10):

$$
\begin{aligned}
u_c{}^2(l_i) = {} & u^2(rep) + u^2(res) + u^2(m) + u^2(cal) + (-l_p\theta_i)^2 u^2(\alpha_i) \\
& + (-l_p\alpha_i)^2 u^2(\theta_i) + (l_p.\theta_p)^2 u^2(\alpha_p) + (l_p.\alpha_p)^2 u^2(\theta_p)
\end{aligned}
\quad (10)
$$

Lastly, the combined standard uncertainty was expanded as $(U)$, defining an interval around $l_i$ known as the coverage interval that is expected to encompass a large fraction of values that could reasonably be attributed to the grating period length $l_i$. The expanded uncertainty is obtained by multiplying $u_c(l_i)$ by a coverage factor k as follows: $U = u_c(l_i) \times k$. Coverage interval $l_i \pm U$ ensures that 95.45% of the expected values of $l_i$ are within the interval.

## 3. Results

The first step was to ensure that the suggested system was properly calibrated and adjusted, keeping in mind that this system was automated using commercial software LABVIEW (National Instruments). Gratings were created in the ablation regime for this purpose. The gratings were produced with 100 periods, a total length of 62 mm, and an exposure time of 300 ms. The $CO_2$ laser was configured with a power equal to 13 W and a frequency of 5 kHz. This frequency was chosen to allow for glass to absorb the energy of the $CO_2$ laser [12,33]. As a result, a high contrast was obtained to aid in the visibility of the grooves under Quick Scope (Mitutoyo, QS-L2010Z/AFC).

After inscribing the diffraction gratings on optical fibers, they were imaged with digital Quick Scope (Mitutoyo, QS-L2010Z/AFC) provided by Colombia's National Institute of Metrology (INM). Figure 5 shows images acquired with Quick Scope to validate the grating period, $S_d$, and $S_w$. Due to the high-frequency modulation of the $CO_2$ laser pulses, the LPFG did not show any deformation or microcurvatures even when high power was used. To measure the mentioned parameters, we extracted the grating period shown in Figure 5a, while the engraved spot surface width and penetration depth were extracted as shown in Figure 5b, respectively. To perform these measurements, ten LPFGs were fabricated, and a total of 80 engraved spots were measured and analyzed (20 spots per grating). The results from the optical characterization are summarized in Table 1. The obtained results show small difference between the desired period (620 μm) and the obtained period (619 μm). This allowed for us to ensure that the system built for LPFG manufacturing had high precision.

**Table 1.** Results of laser penetration measurement.

| Programmed Value (μm) | $l_i$ (μm) | Error (μm) | U (μm) |
|---|---|---|---|
| 620.0 | 619.0 | −1.0 | 4.0 |

From the measurement of the period length of the recorded LPFGs, feedback could be sent to the program code designed in Labview to adjust the lengths in which the linear station moved in the engraving process. As a result, the recording error was $-1\ \mu$m (see Table 1), implying that a correction was needed for each linear station movement to move the period closer to 620 $\mu$m On the other hand, to ensure reproducibility when recording the LPGs according to the model presented in Equation (7), the temperature had to be at 20 °C $\pm$ 1 °C, since the thermal expansion coefficient associated with the linear displacement station can cause a length variation of 1 micrometer per degree Celsius for the total distance of the LPFG ($\triangle$L = $\alpha$L$\theta$ = 10.5$\times$10$^{-6}$/°C $\times$ 62 mm = 1 $\mu$m/°C).

In the same way, the obtained results for the laser spot depth ($S_d$) and spot width ($S_w$) are presented in Table 2. The depth was very close to the location of the fiber's core during the entire displacement of the stage, meaning that the setup for optical alignment promises good repeatability of LPFG fabrication.

**Table 2.** Results of laser spot measurement.

| Geometry Parameter | Mean Value (µm) | U (µm) |
|:---:|:---:|:---:|
| $S_d$ | 49.4 | 1.2 |
| $S_w$ | 50.3 | 3.0 |

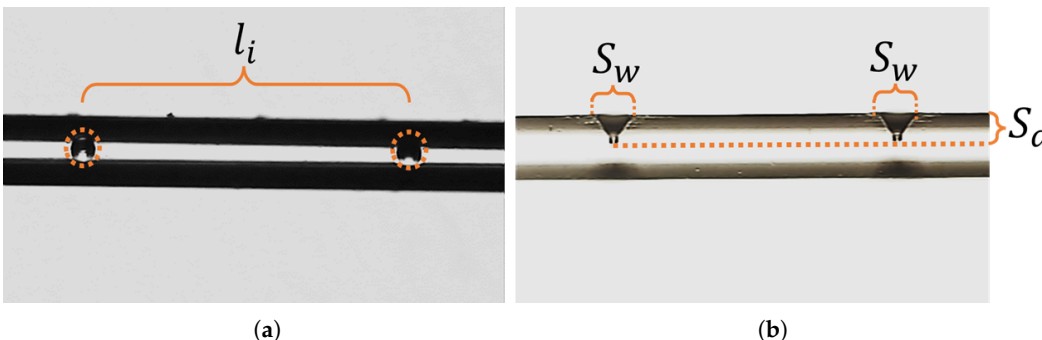

(**a**)                                                                           (**b**)

**Figure 5.** (**a**) Measurement points for fiber gratings; the grating period was extracted from the focal plane observed. (**b**) The engraved spot surface width and penetration depth were extracted.

The LPFG recording system was used to manufacture 10 LPFGs with the following parameters: $\wedge$ of 100 $\mu$m, total length of 62 mm, and exposure time of 300 ms. To carry out this fabrication, the $CO_2$ laser was configured with a power equal to 5 W and a frequency of 5 kHz. The obtained spectrum of one of the constructed LPFG is shown in Figure 6, showing the three low-order cladding modes of $LP_{02}$, $LP_{03}$, $LP_{04}$ at 1525, 1541, and 1555 nm, respectively. The spectra of the 10 LPFGs were analyzed, and a standard deviation of less than 5 nm was found for the three resonance peaks, indicating that the LPFGs had good repeatability.

FIMMWAVE® (Photon Design Inc. (Oxford City, UK)) software based on the finite difference method (FDM) is a precise computational tool for analyzing complex optical waveguides. It was used to simulate an LPFG with a period of 620 $\mu$m and a total length of 62 mm, inscribed on a standard optical fiber SMF28e. Figure 7 shows the transmission spectrum of this LPFG from 1500 to 1580 nm, showing three resonant peaks at 1526.5, 1539, and 1561 nm, which were quite close to those found experimentally. Likewise, the calculated modal fields at these wavelengths confirm that they corresponded to the $LP_{02}$, $LP_{03}$, $LP_{04}$ (see insets in Figure 7).

As mentioned in Section 2, an LPFG's resonance peaks are sensitive to changes in the environment surrounding the fiber cladding. So, the next sections detail the behavior of the created LPFGs to changes in temperature and strain.

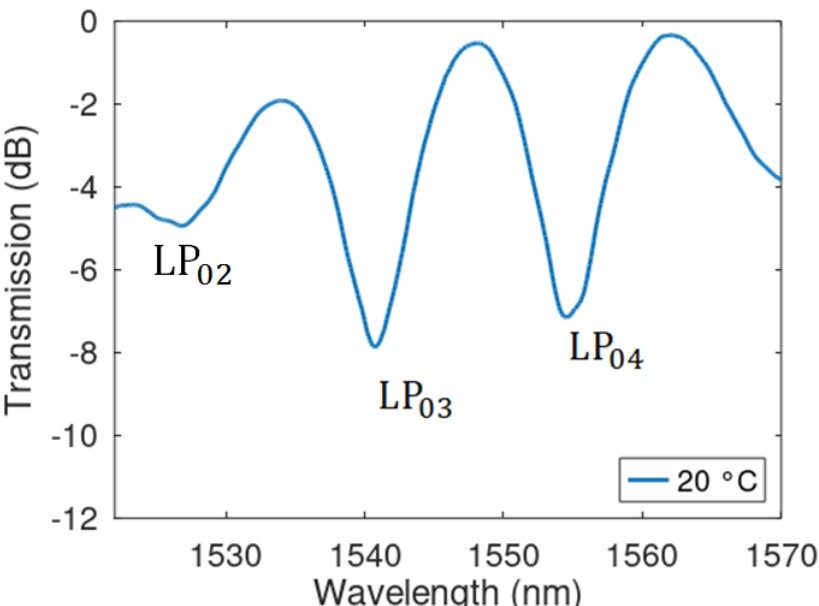

**Figure 6.** Spectral response of the fabricated LPFG. The spectra show the three low−order cladding modes $LP_{02}$, $LP_{03}$, $LP_{04}$ at 1525, 1541, and 1555 nm, respectively.

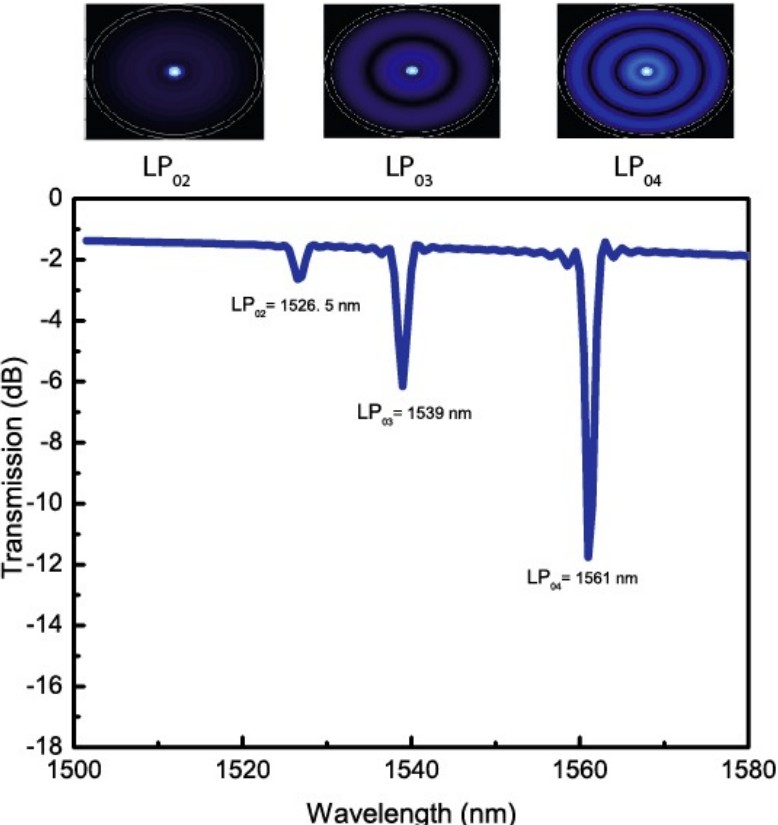

**Figure 7.** Numerical initial transmission spectrum for an LPFG of a period of 620 µm on an SMF28e.

## 4. Discussion

### 4.1. Thermal Characterization

The experimental setup illustrated in Figure 8 was used to evaluate the spectral response of the LPFG to thermal changes. A ceramic Peltier module (12V- 4A) was placed

beneath the fiber optic. The temperature of the Peltier module was controlled with a power and current source that enabled thermal stability over each temperature measurement point. Likewise, a SLED optical source and an optical spectrum analyzer (Yokogawa, AQ-6370B) were used to examine the spectral response when the temperature was increased from 20 to 97 °C, while the LPFG was immobilized to minimize fluctuations in the resonance peaks caused by strain, torsion, or curvatures.

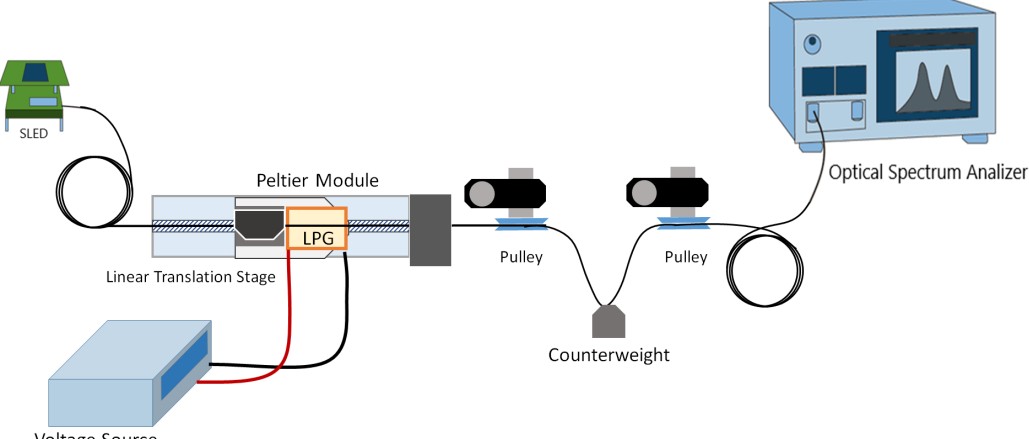

**Figure 8.** Experimental setup used for temperature characterization of engraved LPFGs.

Figure 9 shows the response of the LPFG when the temperature changed from 10 to 97 °C. It is evident that modes $LP_{03}$ and $LP_{04}$ presented a shift towards longer wavelengths. This was a consequence of variations induced by thermal changes that modified the phase matching condition between the core mode and the cladding modes. Likewise, this also affected the peak amplitude because the coupling efficiency was also affected. On the other hand, the sensitivity to thermal changes of both modes was evaluated. Figure 10a,b represent the linear regression for the data obtained from cladding modes $LP_{03}$ and $LP_{04}$ at different thermal points, respectively. The black and red dotted lines represent the coverage interval for the fitted curve and the experimental data, respectively. Figure 10a,b show that the sensitivities for the cladding modes $LP_{03}$ and $LP_{04}$ were 56 and 58 pm/°C, respectively. Despite mode $LP_{04}$ exhibiting a higher sensitivity than $LP_{03}$ did, this mode also had high dispersion and less repeatability. Because of the above, for further applications as a temperature sensor, it is recommended to use cladding mode $LP_{03}$ due to its repeatability and stability in shape. Likewise, it is evident that, in both cases, the LPFG-based sensor presented high linearity, which is desirable in industrial applications.

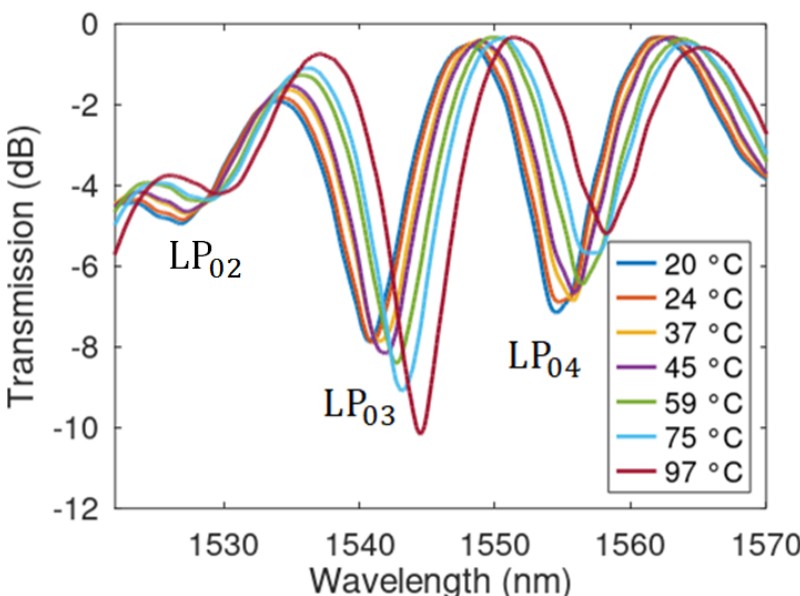

**Figure 9.** Transmission spectrum of the fabricated LPFG at different temperatures.

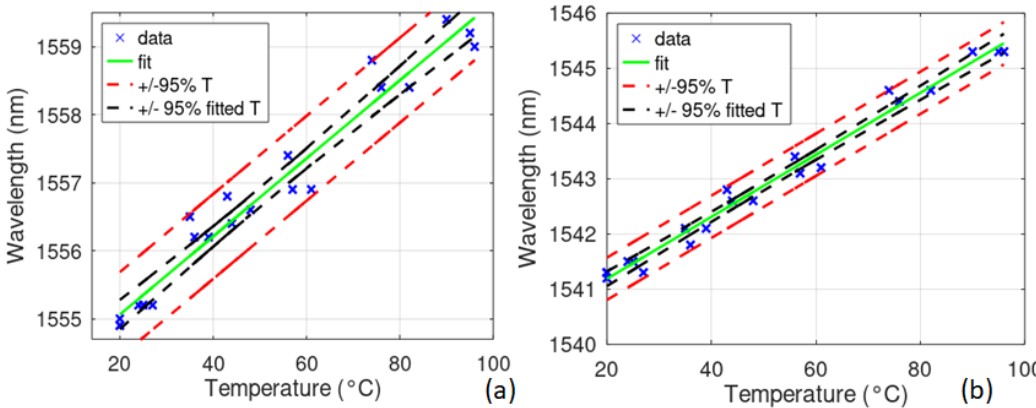

**Figure 10.** (**a**) Temperature response of the $LP_{03}$ mode of the LPFG as a function of thermal changes. (**b**) Temperature response of the $LP_{04}$ mode of the LPFG as a function of thermal changes.

### 4.2. Strain Characterization

Lastly, the LPFG was studied when exposed to axial stress variations. An experimental setup, as shown in Figure 11, was used to carry out this experiment. The following are the significant changes between this arrangement and the one shown in Figure 9. The device was placed inside a polystyrene box to keep the sensor thermally insulated, and weights were utilized to change the axial deformation applied to the LPFGs. A model similar to that proposed by Black et al. in [34] was employed to compute the microstrain imposed on the optical fiber. In that model, the authors suggested calculating microstrain as a function of the Young's modulus of silica ($E_{silica}$), the applied force (F), and the cross-sectional area of the optical fiber ($A_{fiber}$).

Figure 12a depicts the transmission spectra following the cladding mode $LP_{03}$ at various axial strain values. These strain fluctuations changed the tensile stress values on the fibers, resulting in minor alterations in the resonance wavelength. The results are shown in Figure 12b, where the sensitivity was 43 pm/$\mu\varepsilon$, which is a bit lower than the average sensitivity attained by this kind of LPFGs, but not atypical [21,23]. The confidence intervals for the fitted curve and the experimental data are represented by black and red dotted lines, respectively.

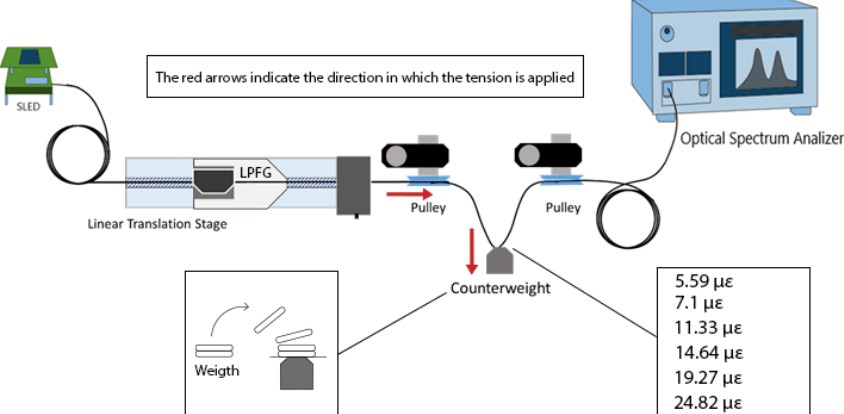

**Figure 11.** Experimental setup used for strain characterization of engraved LPFGs.

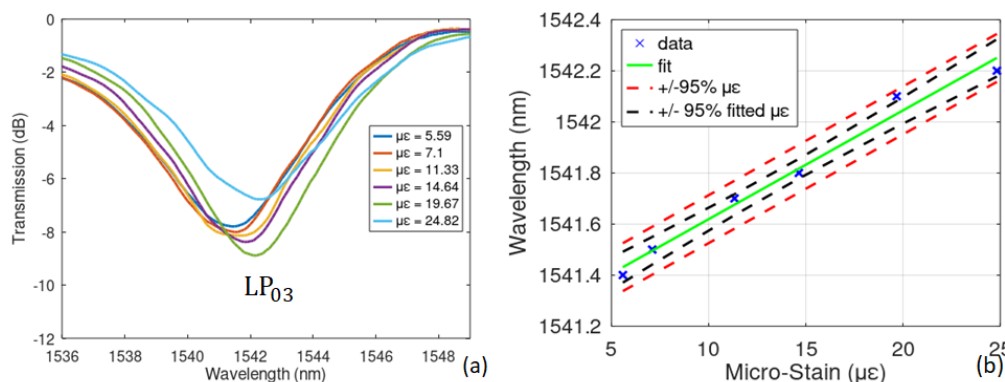

**Figure 12.** (**a**) Spectral response to a microstrain test. (**b**) The wavelength shift of the $LP_{03}$ mode.

## 5. Conclusions

In this paper, an engraving setup was characterized in order to ensure the repeatability and reproducibility requirements for the manufacture of LPFGs in standard optical fibers. A $CO_2$ laser was effectively employed for this purpose. Furthermore, for the first time, an uncertainty model of the engraving system based on point-to-point $CO_2$ laser exposure was presented and implemented.

On the other hand, the physical properties of the produced LPFGs were tested using a Mitutoyo Quickscope, which allowed for us to confirm the LPFG pitch, engraved spot surface width, and penetration depth. The measured period was $618.7 \pm 4.0$ μm with an error of 1.3 μm. The width and depth of each engraving spot in material ablation was used to check the uniformity of the laser damage caused on the fiber, with mean values of $50.3 \pm 3.0$ and $49.4 \pm 1.2$ μm. Thus, the uncertainty model evidenced the good control of the experiment.

Lastly, the LPFGs were used as strain and temperature sensors. Thus, when cladding modes $LP_{03}$ and $LP_{04}$ were investigated, the findings showed that the constructed LPFGs could be utilized to sense temperature changes between 20 and 97 °C with sensitivities of 56 and 58 pm/°C, respectively. Additionally, experimental validation demonstrated that the capacity these sensors to sense strain was good. When the applied axial tension was adjusted from 5.6 to 25 μ$\varepsilon$, the sensor displayed a sensitivity of 43 pm/μ$\varepsilon$.

As future work, we aim to use the results of this work for a closed-loop engraving technique to construct LPFGs on nonstandard optical fibers such as few-mode optical fibers, thereby generating modal conversion systems. The suggested system can also be used for

other postprocessing techniques that allow for us to construct various types of sensors, such as small interferometers and resonant ring structures.

**Author Contributions:** Conceptualization, investigation, methodology, formal analysis, and writing—original draft: E.R.-V. and N.G.-C. Optical system setup, data curation, and operation: J.P.M. and S.V.-G. Measurements: J.G.-A. Analysis, data curation, and writing: E.R.-V., J.G.-A., J.P.M. and S.V.-G. All authors have read and agreed to the published version of the manuscript.

**Funding:** Funding for this project came from the *Instituto Tecnologico Metropolitano* University as project P21101.

**Institutional Review Board Statement:** Not applicable.

**Informed Consent Statement:** Not applicable.

**Data Availability Statement:** Not applicable.

**Acknowledgments:** The authors acknowledge the support of *Instituto Tecnologico Metropolitano* through project P21101.

**Conflicts of Interest:** The authors declare to have no conflict of interest.

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
