# Peer review of "Metrological Characterization of a CO2 Laser-Based System for Inscribing Long-Period Gratings in Optical Fibers"

_instruments, doi:10.3390/instruments6040079_

Round 1

Reviewer 1 Report

The authors proposed a CO2 laser-based system for fabrication of long period fiber gratings. To analyze the mechanical and thermal effects, a model was proposed for optimizing the fabrication. It looks the results are good. However, the strain measuring range is much smaller which will limits the sensing performance. Why did the authors use a high power to laser machining the fiber as shown in Fig. 5? Typically, a smaller power is enough to fabricate LPGs with large attenuation peaks. The thermal effect induced LPG could be not good as the refractive index-induced LPG. The authors should the comparable results. Additionally, the transmission is also related to the period number, and this result should be presented as well.

Author Response

Responses to each comment are presented in the attached file.

Reviewer 2 Report

The manuscript presents a CO2 laser-based fabrication of long period gratings in standard SMF fiber. The performance metrics of the modified fiber structure has been tested against temperature and external tensile-stress induced strain sensing. Looks like the research group is establishing an LPG fabrication set-up in the lab but the novelty is not clear enough for the results to be published in a scientific journal. There are similar set-ups and mechanism of LPFG fabrication already reported in the literatures. The sensing performance of the fabricated sensors are even lower than the previously reported values.

The authors are emphasizing on the higher reproducibility and repeatability of the proposed technique in fabricating LPG sensors, but the experiments are not carried out to test the reproducibility and repeatability of the sensor fabrications methods. I suggest more sensors fabrication to test the process capability of the fabrication technique.

The novelty claimed here “creation of a mathematical model for the characterization of the LPFG manufacturing process using the CO2 laser-based technique” is not only vague and unclear, but also insignificant, if it is for just measuring in-situ grating period/length. I suggest authors focus on how this model is used to control the fabrication process for reproducibility and repeatability. For example, clarify whether the calculation from the model is utilized in some feedback control of LPG fabrication with intended period and length. See the example case of machine vision approach in LabVIEW in controlling the diameter of the single crystal fiber fabrication below.

"Machine vision approach of process control during single crystal fiber growth via laser heated pedestal growth method," Proc. SPIE 12105, Fiber Optic Sensors and Applications XVIII, 1210508 (27 May 2022); https://doi.org/10.1117/12.2618538.

Some more comments/suggestions

1. There are many instances the authors have cited some explanations/results/scientific reasons without citing the references. Please cite source literatures.

 Few examples where citations are required to support the statements.

 In line 141-143,

“Different sensors based on LPFGs with low-order cladding modes (fabricated using SMF-28 optical fiber) had reported temperature sensitivities between 30 pm/°C and 100 pm/°C”. Provide reference.

In line 213-216,

“The CO2 laser was configured with a power equal to 13 W, and a frequency of 5 kHz.

This frequency was choice due that it is probably the best frequency for glass to absorb the energy of a CO2 laser.” Provide reference.

In line 290-291,

“The low sensitivity to axial strain is due to the use of high frequencies to engrave the LPFGs, resulting in negligible physical effects on the fiber.” Provide reference.

2.     In Fig. 6,

 The transmission is 100 % for the wavelengths between two LPG resonance dips. This is likely due to inaccurate measurement or the calculation error. There should be always some insertion loss, no matter how low it is. Please explain this artifact and explain how did you calculate this transmission value in dB?  See the LPG transmission spectrum reported by others for comparison.

Moreover, how do you know the dips in the transmission spectrum correspond to LP02, LP03, and LP04 and not LP00, LP01, LP02 etc..

3.     Include the detail of the optical components in lens array that is used for focal plane control and explain how you control the diameter of the CO2 laser beam.

4.     Briefly explain how you calculate the micro-strain applied in fiber when you hang different mass or provide reference.

5.     In the conclusion section, authors claim “Thus, the uncertainty model evidence a good control of the experiment.” This statement is what the novelty is claimed for, but no justification has been offered in the manuscript. More focus has to be given to prove this claim, clarify how, and justify it.

Author Response

(The authors gave the same response as above.)

Round 2

Reviewer 2 Report

The authors have done a good job in addressing the reviewers comments and concerns. As a result, I believe,  the manuscript has been improved in many aspects from clarification of the novelty to presentation of the results and analysis more distinctly. The fabrication approach aided by the uncertainty mathematical model offers some merits towards the process optimization in fabricating LPGs using CO2 laser. Thus the manuscript can be considered for publication in instruments.